# Development of a Polymeric Pharmacological Nanocarrier System as a Potential Therapy for Spinocerebellar Ataxia Type 7

**DOI:** 10.3390/cells12232735

**Published:** 2023-11-30

**Authors:** Fabiola V. Borbolla-Jiménez, Ian A. García-Aguirre, María Luisa Del Prado-Audelo, Oscar Hernández-Hernández, Bulmaro Cisneros, Gerardo Leyva-Gómez, Jonathan J. Magaña

**Affiliations:** 1Laboratorio de Medicina Genómica, Departamento de Genética (CENIAQ), Instituto Nacional de Rehabilitación-Luis Guillermo Ibarra Ibarra (INR-LGII), Ciudad de México 14389, Mexico; fabiola_borbolla@comunidad.unam.mx (F.V.B.-J.); ohernandez@inr.gob.mx (O.H.-H.); 2Programa de Ciencias Biomédicas, Facultad de Medicina, Universidad Nacional Autónoma de México (UNAM), Ciudad de México 04510, Mexico; 3Departamento de Bioingeniería, Escuela de Ingeniería y Ciencias, Tecnologico de Monterrey, Ciudad de México 14380, Mexico; ian.garcia@tec.mx (I.A.G.-A.); luisa.delprado@tec.mx (M.L.D.P.-A.); 4Departamento de Genética y Biología Molecular, Centro de Investigación y de Estudios Avanzados (CINVESTAV-IPN), Ciudad de México 07360, Mexico; bcisnero@cinvestav.mx; 5Departamento de Farmacia, Facultad de Química, Universidad Nacional Autónoma de México (UNAM), Ciudad Universitaria, Ciudad de México 04510, Mexico

**Keywords:** nanotechnology, pharmacological treatment, polymeric nanocarrier, polyQ aggregation, rapamycin, spinocerebellar ataxia type 7

## Abstract

Spinocerebellar ataxia type 7 (SCA7) is an autosomal-dominant inherited disease characterized by progressive ataxia and retinal degeneration. SCA7 belongs to a group of neurodegenerative diseases caused by an expanded CAG repeat in the disease-causing gene, resulting in aberrant polyglutamine (polyQ) protein synthesis. PolyQ ataxin-7 is prone to aggregate in intracellular inclusions, perturbing cellular processes leading to neuronal death in specific regions of the central nervous system (CNS). Currently, there is no treatment for SCA7; however, a promising approach successfully applied to other polyQ diseases involves the clearance of polyQ protein aggregates through pharmacological activation of autophagy. Nonetheless, the blood–brain barrier (BBB) poses a challenge for delivering drugs to the CNS, limiting treatment effectiveness. This study aimed to develop a polymeric nanocarrier system to deliver therapeutic agents across the BBB into the CNS. We prepared poly(lactic-co-glycolic acid) nanoparticles (NPs) modified with Poloxamer188 and loaded with rapamycin to enable NPs to activate autophagy. We demonstrated that these rapamycin-loaded NPs were successfully taken up by neuronal and glial cells, demonstrating high biocompatibility without adverse effects. Remarkably, rapamycin-loaded NPs effectively cleared mutant ataxin-7 aggregates in a SCA7 glial cell model, highlighting their potential as a therapeutic approach to fight SCA7 and other polyQ diseases.

## 1. Introduction

Spinocerebellar ataxia type 7 (SCA7) is an autosomal dominant neurodegenerative disease characterized by progressive cerebellar ataxia and visual impairment, due to cone-rod and retinal dysfunction [1,2]. SCA7 is one of the five most common ataxias worldwide [3] and has been identified in diverse ethnic backgrounds [4]; however, it shows high incidence in specific countries, including Sweden, Finland, South Africa, Zambia, and Mexico [5,6,7]. The mutation causing SCA7 is an abnormal expansion of a CAG triplet repeat located in the *ATXN7* gene coding region, on the chromosomal region 3p12-21.1 [8]; the expanded CAG repeats lead the translation of ataxin-7 carrying an abnormally large glutamine-rich tract (polyQ). Mutant ataxin-7 undergoes aberrant folding due to the formation of β-sheet structures [9,10], which results in the accumulation of protein aggregates within neurons and glial cells [11,12,13,14]. In consequence, mutant ataxin-7 acquires a dominant toxic function that provokes diverse cellular alterations, including bioenergetic defects, oxidative stress, and mitochondrial dysfunction [11,15,16]. At the molecular level, aggregates of mutant ataxin-7 sequester ubiquitin proteasome subunit 20S, chaperones Hsp70 and Hsp40, caspase-3, transcription factor CREB-binding protein (CBP) and p53, as shown in cellular and mouse models of SCA7, and in brain and retina postmortem tissues derived from SCA7 patients [17,18,19]. Furthermore, nuclear aggregates of ataxin-7 hijack components of the Spt-Ada-Gcn5-acetyltransferase (SAGA) chromatin modifier complex, including ubiquitin specific peptidase 22 (USP22) and wild type ataxin-7 itself, which ultimately affects the function of a plethora of genes involved in critical cellular pathways, including autophagy and the ubiquitin-proteasome system [20]. At the systemic level, this gross dysregulation drives different regions of the central nervous system (CNS) to degenerate, including cerebellum and its associated structures (inferior olive, cerebellar cortex, dentate nucleus, and pontine nucleus), brainstem and retina (cone/rod photoreceptors, ganglion, and bipolar cells) [21,22,23,24].

Owing to the central role of mutant ataxin-7 in the pathophysiology of SCA7, the current therapeutic approaches are mainly aimed to suppress the mutant protein expression at transcriptional level using RNA interference and antisense oligonucleotides [25], or to proteolytically eliminate mutant ataxin-7 aggregates using drugs with the ability to activate the ubiquitin–proteasome and/or autophagy systems [26]. However, the release of therapeutic agents into the CNS is a major challenge because the blood–brain barrier (BBB), a membranous wall formed by endothelial cells connected by tight junctions and surrounded by astrocyte feet, greatly limits their penetration into brain interstitium [27,28]. Favorably, nanotechnology has provided new drug delivering systems with the capacity to reach the CNS and extend the therapeutic efficacy of drugs [25,28]. Specially, nano-sized carriers composed of biocompatible and biodegradable materials facilitate the passage of drugs through the BBB, while providing proper biodistribution, controlled release, and low toxicity of therapeutic drugs [29,30,31,32]. In this study, we developed a nanocarrier system with specific physicochemical and pharmacological properties for SCA7 therapy, which was composed of poly lactic-co-glycolic acid (PLGA)-based nanoparticles (NPs) modified with poloxamer188 and loaded with rapamycin (autophagy inducer). The therapeutic potential of rapamycin-loaded NPs was evidenced by their ability to clear mutant ataxin-7 aggregates in a SCA7 glial cell model by inducing autophagy.

## 2. Materials and Methods

### 2.1. Materials

Reagents for the elaboration of NPs, including PLGA (Poli (D,L-lactide-co-glycolide, 50:50 mol wt 30,000–60,000)), PVA (Polyvinyl alcohol: Mowiol^®^, PM: ~31,000), Poloxamer 188 (Pluronic^®^ F-68), coumarin-6 and rapamycin, were purchased from Sigma Aldrich^®^ (Merck KGaA, Darmstadt, Germany). Mannitol was obtained from Droguería Cosmopolita (Mexico City, Mexico), while ethyl acetate was acquired from Sprectrum^®^ (Spectrum Laboratory Products, New Brunswick, NJ, USA). Acetonitrile was kindly donated by Bernad-Bernad MJ. Dulbecco’s Modified Eagle Medium (DMEM), DMEM/F12, Minimum Essential Medium (MEM), penicillin-streptomycin, trypsin-EDTA, and PBS were obtained from Gibco^®^/Life Technologies (ThermoFisher Scientific, Waltham, MA, USA); Fetal Bovine Serum (FBS) and FBS tetracycline-free were obtained from Biowest^®^ (Labclinics, Barcelona, Spain) Reagents for biological assays, including trypan blue, paraformaldehyde, and P6148, were supplied by Sigma Aldrich^®^ (Merck KGaA, Darmstadt, Germany). The Cell Proliferation Kit used for MTT (3-[4,5-dimethylthiazol-2-yl]-2,5 diphenyl tetrazolium bromide) assays was purchased from Roche^®^ (Roche Diagnostics GmbH, Mannheim, Germany). Vectashield^®^ Antifade Mounting Medium with DAPI was supplied by Vector Laboratories (Maravai LifeSciences, San Diego, CA, USA), while rhodamine-phalloidin was obtained from Jackson ImmunoResearch Laboratories (West Grove, PA, USA). The Annexin V & Dead Cell kit used for the evaluation of early and late apoptosis was supplied by Muse^®^ (Luminex; Austin, TX, USA). For Western blotting experiments, antibodies anti-caspase-3 (ab13585), anti-caspase-9 (ab202068), and anti-ataxin7 (ab95013) were acquired from Abcam (Boston, MA, USA), while anti-actin antibody was a gift from Dr. Manuel Hernández (Cinvestav, Mexico City, Mexico). Staurosporine was supplied by Sigma-Aldrich^®^ (Merck KGaA, Darmstadt, Germany), while FL-1000-FITC and anti-LC3 antibody were obtained from Medical & Biological Laboratories (Nagoya, Japan).

### 2.2. Preparation of NPs

PLGA NPs with PVA and Poloxamer 188 NPs (hereafter called PLGA NPs) were elaborated by the emulsion-diffusion procedure. Briefly, double-deionized water-saturated with ethyl acetate (1:1) was obtained and the organic phase (OP) and the aqueous phase (AP) were separated and stored. Next, PLGA (108 mg) was dissolved in 10 mL OP, while PVA (500 mg) and poloxamer 188 (30 mg) were dissolved in 20 mL AP. Afterwards, AP was mixed with OP and the resultant dispersion was emulsified in a high-speed homogenizer (Ultra Turrax T18, IKA^®^, Charlotte, NC, USA) at 11,000 rpm for 10 min. Then, an excess of double-deionized water (266%) was added to promote diffusion of the solvent and NPs formation. NPs were transferred to a ball funnel to remove the ethyl acetate in a rotary evaporator (Heidolph^®^ Instruments GmbH & Co., KG, Schwabach, Germany) at 60 rpm for 1 h and 40 °C. Then, NPs were purified in a centrifuge at 11,000 rpm for 90 min at 4 °C (Model 5804 R Centrifuge; Eppendorf^®^, Hamburg, Germany); the supernatant was discarded, and the pellet was gently washed 3 times with 1 mL double-deionized water. Then, the pellet was resuspended in double-deionized water, and mannitol (15.6 mg/mL) was added as cryoprotectant. The resultant dispersion was maintained on a magnetic stirrer for 1 h. Finally, NPs were frozen for 24 h and lyophilized at 120 × 10^−3^ Mbar for 36 h with a collector temperature of −48 °C. All the steps were carried out in an aseptic environment. After lyophilization, the freeze-dried NPs were sterilized with UV light for 2 h at 100 μJ/cm^2^ of intensity, as previously reported [33]. PLGA/PVA/Plx188/Rap NPs (hereafter called Rap-PLGA NPs) were elaborated following the above-described procedure with the exception that rapamycin (5 mg) was added to the PLGA-containing OP and the produced NPs were lyophilized and stored at 4 °C protected from light. 

### 2.3. Physicochemical Characterization

#### 2.3.1. Scanning Electron Microscope (SEM)

NPs morphology was determined using a Scanning Electron Microscope (FIB-SEM Crossbeam 550; Zeiss^®^; Jena, Germany). A 3.5 μL aliquot (1 mg/mL) of previously air-dried PLGA NPs or Rap-PLGA NPs placed on a glass was sputtered with gold in a JEOL fine coat Ion Sputter JFC-1100 (JEOL, Tokyo, Japan), and representative images were captured.

#### 2.3.2. Particle Size, PDI and Z-Potential Measurement

For the determination of the particle size, polydispersity index (PDI) and Z-potential parameters, a Zetasizer Nano (Malvern Instrument ZS90; Malvern, UK) was used. Particle size was determined at 25 °C using Dynamic Light Scattering (DLS), while Z-potential was calculated at 25 °C using an electrophoretic mobility Velocimetry Laser Doppler. All measurements were performed 10–15 times and the results are shown as the average value ± standard deviation (SD).

#### 2.3.3. Fourier Transform-Infrared Spectroscopy (FTIR)

FTIR spectra of freeze-dried PLGA NPs and Rap-PLGA NPs (lyophilized without mannitol), as well as raw reagents (PLGA, PVA, Plx188, and rapamycin) were recorded at the range of 4000–400 cm^−1^ using an FTIR Nicolet 6700 (Thermo Fisher Scientific, Waltham, MA, USA).

#### 2.3.4. Differential Scanning Calorimetry (DSC)

The thermal properties of freeze-dried PLGA NPs and Rap-PLGA NPs (lyophilized without mannitol) and raw reagents (PLGA, PVA, Plx188, and rapamycin) were assessed using DSC (DSC 2910, Modulated TA Instruments, New Castle, DE, USA). Samples were placed into hermetic aluminum cells; then they were heated from 26 °C to 250 °C at a range of 10 °C/min under a constant nitrogen atmosphere.

#### 2.3.5. Thermogravimetric Analysis (TGA)

The thermal stability of freeze-dried PLGA NPs and Rap-PLGA NPs (lyophilized without mannitol) as well as raw materials (PLGA, PVA, Plx188, and rapamycin) was measured using a HiRes TGA 2950 Thermogravimetric analyzer (TA Instruments, New Castle, DE, USA). Samples were heated from 35 °C to 500 °C at a range of 10 °C/min under constant nitrogen atmosphere.

#### 2.3.6. Entrapment Efficiency and Drug Loading

Entrapment efficiency (EE) and drug loading (DL) of rapamycin were determined using the Nanodrop 2000^®^ (Thermo Scientific, Wilmington, DE, USA). A fresh batch of Rap-PLGA NPs was centrifugated and the pellet was washed 3 times with double-deionized water and air-dried for 18 h. Then, the pellet was broken with acetonitrile during 72 h at 4 °C on a magnetic stirrer, for extracting the loaded rapamycin. The obtained suspension was centrifuged at 3000 rpm for 10 min and filtrated on 0.22 nm cellulose membrane filter (Millipore^®^; Burlington, MA, USA). Finally, the samples were recorded at 279 nm using the Nanodrop and quantified using a calibration curve previously prepared. The EE and DL were calculated according to the following equations:(1)$$\mathrm{EE}\left(\%\right)=\left(\frac{\mathrm{Amount}\;\mathrm{of}\;\mathrm{rapamycin}\;\mathrm{in}\;\mathrm{NPs}}{\mathrm{Initial}\;\mathrm{amount}\;\mathrm{of}\;\mathrm{rapamycin}}\right)*100$$
(2)$$\mathrm{DL}\left(\%\right)=\left(\frac{\mathrm{Amount}\;\mathrm{of}\;\mathrm{rapamycin}\;\mathrm{in}\;\mathrm{NPs}}{\mathrm{Amount}\;\mathrm{of}\;\mathrm{NPs}\;\mathrm{with}\;\mathrm{rapamycin}}\right)*100$$

#### 2.3.7. Determination of Rapamycin Release Kinetics from NPs

The release profile of rapamycin from NPs was analyzed by microdialysis methodology. Briefly, dialysis tubing cellulose membrane (MWCO: 12–14 kDa with 45 μm diameter pore, Sigma-Aldrich^®^; Merck KGaA, Darmstadt, Germany) was hydrated for 24 h using double deionized water. Next, 60 mg of Rap-PLGA NPs were dissolved in PBS 1X to obtain a final concentration of 40 mg/mL, and the dissolution was placed inside a sealed dialysis bag. Afterwards, the dialysis bag was introduced into an amber bottle containing 10 mL release medium (PBS 20% *v*/*v* acetonitrile), with constant magnetic stirring at 35 °C. At predetermined time points, the dissolution medium (10 μL) was withdrawn and replaced with an equal volume of fresh medium. For drug content determination, the sample solution was registered at 279 nm (*n* = 3) in the Nanodrop, using a calibration curve previously prepared. For the concentration of NPs, sink conditions were considered for rapamycin release. Data obtained were adjusted to zero-order, first-order, Higuchi, and Korsmeyer–Peppas mathematical models.

#### 2.3.8. Stability of NPs in Dispersion

To determine the stability of dispersed NPs in storage, lyophilized PLGA NPs were hydrated with double deionized water at a final concentration of 1 mg/mL and stored protected from the light at 4 °C. Every measurement at the indicated time periods included particle size, PDI, and Z-potential.

### 2.4. Biological Characterization

#### 2.4.1. Cell Culture

MIO-M1 retinal glial cells and its derivative cell line that express ataxin-7 in an inducible manner (MIO-M1-Q64 cells) [34], as well as SH-SY5Y human neuroblastoma cells were cultured at 37 °C in a 5% CO_2_ humidified atmosphere. MIO-M1 and MIO-M1-Q64 cells were grown in DMEM, while SH-SY5Y cells were grown in a 1:1 mixture of MEM and DMEM-F12; all media were supplemented with 10% FBS and 100 U/mL penicillin/100 µg/mL streptomycin. Manipulation of inducible model was performed with culture media containing 10% FBS tetracycline-free. Furthermore, MIO-M1-Q64 cell cultures were added with 250 µg/mL G418 and 0.16 µg/mL puromycin (Sigma-Aldrich; Merck KGaA, Darmstadt, Germany). When indicated, these cell cultures were added with 1 µg/mL doxycycline (Dox) and 50 μM chloroquine (Sigma-Aldrich) for 24 h to induce mutant ataxin-7 expression and inhibit autophagy, respectively.

#### 2.4.2. Nanoparticle Uptake

A fresh batch of coumarin-6 (C6)-loaded NPs was prepared to assess cellular uptake. Briefly, MIO-M1 and SH-SY5Y cells were seeded on microscope coverslips at 1 × 10^4^ cell density, prior to be treated with 150 μg/mL PLGA NPs or C6-PLGA NPs (coumarin-6 loaded NPs) for 2 h. Afterwards, the coverslips were washed 3 times with PBS 1X and fixed with 4% paraformaldehyde in PBS for 20 min. Then, they were washed 3 times with PBS 1X and mounted on microscope slides with 8 μL VectaShield antifade medium containing diamino-2-phenylindole (DAPI) (Vector Labs Inc., Burlingame, CA, USA) to be examined on an Eclipse Ti-E inverted confocal laser scanning microscope (NiKon; Tokyo, Japan).

#### 2.4.3. Cell Viability Assays

Cell viability experiments were carried out using a MTT (3-(4,5-dimethylthiazol-2-yl)-2,5-diphenyltetrazolium bromide) kit (Sigma-Aldrich; Merck KGaA, Darmstadt, Germany). Briefly, cells were seeded on a 96-well plate at 1 × 10^4^ cell density/well and cultured for 24 h prior to being washed once with PBS 1X and treated for 24 or 72 h with different concentrations of PLGA NPs or Rap-PLGA NPs (3, 6, 12, 24, 50, 100, and 150 μg/mL). Afterwards, the supernatant was discarded, and the cells were washed twice with PBS 1X and added with 100 μL of growth medium containing 8 μL MTT (10 mg/mL) prior to being incubated at 37 °C with 5% CO_2_ for 4 h. Finally, the formazan crystals were dissolved, and the plate was read at 590 nm in an iMarK Microplate Reader (Bio-Rad^®^; Hercules, CA, USA).

#### 2.4.4. Analysis of Cell Morphology by Confocal Microscopy

MIO-M1 and SH-SY5Y cells were seeded on microscope coverslips at 1 × 10^4^ cell density and cultured for 24 h, prior to be treated with 25 μg/mL or 150 μg/mL of PLGA NPs or Rap-PLGA NPs for 72 h. After treatments, the cells were washed twice with PBS 1X and fixed with 4% paraformaldehyde in PBS for 10 min. Then, they washed twice with PBS and permeabilized with 0.2% triton X-100 in PBS 1X for 5 min. They were washed again and blocked with 0.5% gelatin and 1.5% SFB in PBS at room temperature for 20 min. Afterwards, the microscope coverslips were washed twice with PBS 1X and incubated with rhodamine-phalloidin for 20 min, and washed twice with PBS. Finally, the coverslips were mounted on microscope slides with 8 μL VectaShield-DAPI to be analyzed in an Eclipse Ti-E inverted confocal laser scanning microscope (NiKon; Tokyo, Japan).

Nuclear morphometric analysis (Feret’s diameter and nuclear area) was determined by the analysis of DAPI stains. Nucleus was delimited by using the wand (tracing) tool and then analyzed using the ROI manager tool. All image analysis were determined by using ImageJ 1.46J.

#### 2.4.5. Annexin V Assays

Early and late apoptosis were analyzed using the Annexin V Kit (Muse^®^, Luminex; Austin, TX, USA). Briefly, cells were seeded on 12-well plates at 1 × 10^5^ cell density and cultured for 24 h prior to be treated for 72 h with 25 μg/mL or 150 μg/mL PLGA NPs or Rap-PLGA NPs. In parallel experiments, the cell cultures were treated with 2 μM of staurosporine (Sigma-Aldrich; Merck KGaA, Darmstadt, Germany) as positive control for apoptosis. After treatment, the attached cells were harvested and centrifuged at 1400 rpm for 5 min, the pellet was resuspended in 100 μL growth medium and 100 μL Annexin V kit was added to each cell sample and incubated for 15 min, prior to be read in the cell analyzer Muse^®^ (Luminex; Austin, TX, USA).

#### 2.4.6. Analysis of Caspases Activation

MIO-M1 and SH-SY5Y cells were treated for 72 h with 25 μg/mL or 150 μg/mL of PLGA NPs or Rap-PLGA NPs, and lysates from treated cells were obtained. Aliquots (30 μg) of protein extracts were heated for 10 min at 95 °C, loaded into 10–12.5% SDS-polyacrylamide gels and electrotransferred onto PVDF membranes for 15 min at 10–15 V. The membranes were blocked for 1 h in TBST 1X [100 mM Tris-HCl pH 8.0, 150 mM NaCl, 0.5% (*v*/*v*) Tween-20] with 3% low-fat dried milk and incubated overnight at 4 °C with primary anti-caspase 3 or anti-caspase-9 antibodies at 1:500 dilution, and anti-actin antibody (loading control). The specific protein signal was developed by incubating for 1 h with the corresponding secondary antibodies (1:30, 000) and the enhanced chemiluminescence (ECL™) Western blotting detection system (Bio-Rad; Hercules, CA, USA), according to the manufacturer´s instructions. Cell cultures treated with 1–2 μM staurosporine were used as positive controls for caspases activation.

### 2.5. Therapeutic Effect

#### 2.5.1. NPs Treatment

MIO-M1-Q64 cells cultured as above-mentioned were treated with 25 μg/mL PLGA NPs, Rap-PLGA NPs or 200 nM of rapamycin for 1 or 3 days.

#### 2.5.2. Analysis of PolyQ Ataxin-7 Subcellular Localization and Autophagy Activation by Confocal Microscopy

MIO-M1-Q64 cells seeded on coverslips at 70% confluence were treated with 1 µg/mL doxycycline to induce expression of exogenous ataxin-7, prior to be treated with 25 μg/mL PLGA NPs or Rap-PLGA NPs for 24 h. When indicated, cell cultures were treated with 50 μM chloroquine to inhibit autophagy in the cell cultures previously treated with NPs. Then, the cell samples were prepared for confocal microscopy examination as described above and incubated overnight at 4 °C with anti-ataxin-7 or anti-LC3 primary antibodies, and then, with the corresponding secondary fluorochrome-conjugated antibodies for 1 h at room temperature (Jackson Immunoresearch Laboratories, West Grove, PA, USA). To stain nuclei, cells were incubated for 10 min at room temperature with 1 mg/mL DAPI (Sigma-Aldrich, St. Louis, MO, USA) in PBS. After washing, coverslips were mounted on microscope slides with VectaShield and observed in an Eclipse Ti-E inverted confocal laser scanning microscope, with representative images shown. Subcellular aggregates of ataxin-7 were counted using ImageJ 1.46j software.

### 2.6. Statistical Analysis

When two experimental groups were compared, an unpaired Student’s *t*-test was performed to determine statistical significance (*p* < 0.05). Data were expressed as mean ± standard error of the mean (SEM). When three or more groups were compared, after a normality test, one-way ANOVA or Kruskal–Wallis test was applied, then, Tukey or Dunnett’s as a post hoc test, respectively, was applied. In cases where parts-of-whole analyses, significance was determined by χ^2^-test. The statistical software GraphPad Prism 9.0.2 software (San Diego, CA, USA) was used for calculations.

## 3. Results and Discussion

### 3.1. Design, Physicochemical, and Structural Characterization of Nanocarriers for SCA7 Therapy

We designed a nanocarrier system composed of PLGA-based nanoparticles (NPs), which were stabilized with PVA and modified with poloxamer 188, a triblock copolymer. NPs elaborated with co-polymers like PLGA, polylactic acid (PLA), or poly(ε-caprolactone) (PCL), have been successfully used to internalize drugs across the BBB, with the help of surfactants, including polysorbates or poloxamers [35,36,37,38]. Specifically, poloxamer 188 (Pluronic F-68) has the ability to enhance NPs uptake by brain capillary endothelial cells via receptor-dependent endocytosis while inhibiting the efflux transporters P-gp [37,39]. With the aim of promoting degradation of ataxin-7 aggregates by autophagy, we entrapped rapamycin (inhibitor of the Mammalian target of rapamycin (mTOR) signaling pathway and autophagy inducer) within PLGA NPs to obtain Rap-PLGA NPs, owing to the ability of rapamycin to induce autophagy and thus promoting the clearance of toxic protein aggregates [40,41].

#### 3.1.1. Nanocarriers Morphology

PLGA NPs and Rap-PLGA NPs (empty and rapamycin-loaded NPs, respectively) were elaborated using the emulsification-diffusion method (see Materials and Methods for details). Then, NPs morphology was observed by SEM (Figure 1). NPs showed spherical shape and smooth surface, characteristics that might enable them to flow [31,39,42]. It is thought that NPs with geometric and small structure can escape from both the reticuloendothelial and phagocytic immune cells systems [43].

#### 3.1.2. Physical Characterization of NPs

The size, PDI, and Z-potential parameters of NPs were assessed and found to be similar between empty and rapamycin-loaded NPs (Table 1), which implies that rapamycin did not affect these properties. The size of NPs (~200 nm) was within an appropriate range (100–500 nm) for drug delivery [30,44], because that size facilitates cell uptake and minimizes the risk for them to be endocytosed by the brain capillary cells [45]. A PDI of <0.05 for both NPs systems indicated a narrow and homogeneous particle formulation optimal for biological applications. This PDI value was close to the lowest values reported [46,47]. On the other hand, NPs were negatively charged and showed high Z-potential values that indicate physical stability [33,48,49]. Furthermore, the presence of Poloxamer188 in these formulations prevents the agglomeration of NPs by steric repulsion forces [50,51].

To determine the stability of dispersed NPs in storage, a stability assay was carried out following the US Food and Drug Administration (FDA) guidance. The PLGA NPs were analyzed for a period of 6 months in dispersion, taking measurements at the time periods shown in Table 1. Minor differences in the size of NPs were observed through time that might be due to NPs hydrolysis upon their hydration, swelling, and erosion [52]. Likewise, PDI values showed slight fluctuations through the 6 months, compared to the measurement determined at the beginning of the study. It is likely that the hydrolysis undergone by the large chains of Plx188 made the NPs smaller but more homogenous. With respect to the Z-potential, it was found to decrease at the fourth and sixth month of storage. It is worth mentioning that the desorption of Poloxamer188 in the NPs formulation produces a variation in the Z-potential. Overall, these results imply that the stability of NPs is suitable for further experimentation, they can be stored in a lyophilized state, and remain stable for up to 4 months after reconstitution.

#### 3.1.3. Chemical Characterization of NPs

The chemical structure and thermal characterization of PLGA NPs, Rap-PLGA NPs, and raw materials were determined using Fourier Transform-Infrared Spectroscopy (FTIR), Differential Scanning Calorimetry (DSC), and Thermogravimetric Analysis (TGA) (Figure 2). The FTIR spectra of raw materials, as well as the effect of the interaction between them, is shown in Figure 2A. For PLGA, the FTIR spectrum exhibited characteristic bands at 2944 cm^−1^, 1742 cm^−1^ and 1087 cm^−1^, which correspond to a stretching of C-H, stretching of C=O of ester type, and C-O stretching of aliphatic polyesters, respectively [53,54,55]. PVA showed characteristic stretching vibration peaks at 3299 cm^−1^, assigned to O-H stretching of intramolecular and intermolecular hydrogen bonds, as well as at 2935 cm^−1^, 2896 cm^−1^, 1733 cm^−1^ and 1094 cm^−1^, corresponding to C-H stretching of alkyl groups, CH_2_ stretching, C=O stretching, and stretching vibrations of C-O groups, respectively. Finally, peaks at 1428 cm^−1^ and 834 cm^−1^ were due to bending of C-H_2_ and rocking bands of C-H_2_, respectively [56,57,58]. Plx188 exhibited vibration peaks at 2881 cm^−1^ and 1108 cm^−1^, assigned to C-H stretching and C-O stretching, respectively [30,59]. The FTIR spectrum of rapamycin showed peaks at 2960 cm^−1^, 1626 cm^−1^, and 1374 cm^−1^, corresponding to CH=CH stretch; as well as bands at 2930 cm^−1^ and 1438 cm^1^ assigned to CH_3_ stretching and a band at 991 cm^−1^ that corresponded to an out of plane vibration of C=O [60]. With respect to PLGA NPs and Rap-PLGA NPs, the characteristic peaks of PLGA, PVA, and Poloxamer 188 were present in the FTIR spectra of both formulations; the absence of the characteristic peaks of rapamycin in the latter NPs formulation might indicate that the drug was effectively encapsulated with no drug–polymer chemical interaction.

The thermal characteristics of NPs formulations and raw materials are shown with the DSC in Figure 2B and Appendix A. PLGA showed a melting temperature (Tm) point of 246 °C and a glass transition temperature (Tg) of 51 °C, which were similar with the previous literature [61,62]. The Tm of PVA was 195 °C, while the Tm of Plx188 was 56 °C, both values according to the previous literature [30,33,56,63]. Rapamycin showed two endothermic peaks at 186 °C and 197 °C, which corresponded to its melting point, in agreement with the literature [63,64]. PLGA NPs exhibited a peak at 54 °C, that must correspond to the Tg of PLGA, and a second pattern of peaks at 156 °C and 166 °C, which may correspond to the Tm of PVA and PLGA, respectively. Concerning Rap-PLGA NPs, this formulation exhibited endothermic peaks at 53 °C, 155 °C and 165 °C, which must correspond to the Tg of PLGA, the Tm of PVA and the Tm of PLGA, respectively. Since the characteristic endothermic peak corresponding to the Tm of Poloxamer188 (57 °C) was not found in either of the two formulations, we argue that the thermal transition assigned to the Tg of PLGA (54 °C for PLGA NPs and 53 °C for Rap-PLGA NPs) could dissolve prior to Poloxamer188. It is worth to mention that none of the endothermic melting peaks of rapamycin were observed in the graph of Rap-PLGA NPs, which might be due to an efficient encapsulation and homogenous entrapment of the drug in an amorphous state, which reinforces the result observed in the FTIR [61]. Since rapamycin melts at higher temperatures than PLGA, crystalline rapamycin could have been dissolved in the melted PLGA during the DSC analysis, thus shifting to an amorphous form.

Finally, Figure 2C and Appendix A show the TGA of PLGA NPs, Rap-PLGA NPs, and raw materials. PLGA started to lose weight at 195 °C and its decomposition was completed at 349 °C. This degradation mechanism could be related to a random chain scission of the polymer due to hydrolysis, as previously reported [33,54]. PVA showed two decomposition stages, ranging from 257 °C to 365 °C and from 380 °C to 468 °C, which agrees with the literature [33]. Poloxamer 188 exhibited a weight loss between 250 °C and 363 °C, which was consistent with the previous literature [30]. Rapamycin showed a decomposition pattern ranging from 196 °C to 460 °C with a sudden thermal change between 220 °C and 250 °C, as previously reported [60]. On the other hand, PLGA NPs initiated to decompose from 216 °C to 329 °C while Rap-PLGA NPs started to decompose at 216 °C and ended at 326 °C. As expected, the TGA decomposition profile in both formulations showed a faster and narrower degradation, which may be due to their large contact surface that make them more reactive to temperature.

#### 3.1.4. Entrapment Efficiency and Drug Loading

The entrapment efficiency (EE) and drug loading (DL) percentages were calculated to determine the percentage of rapamycin entrapped in Rap-PLGA NPs and the amount of rapamycin loaded per weight unit of the NPs, respectively. Using a calibration curve for the quantifying of rapamycin, we obtained EE and DL values of 43.64% and 0.95% (8.43 ng Rap/μg of NPs). These data support the conclusion that the manufacturing method applied in this work was sufficiently effective. Alternative manufacturing systems, such as liposomes or water-soluble drug encapsulation, may have higher EEs and DLs; nonetheless, the encapsulated drug tends to leak, affecting its sustained release [65,66,67]. Of note, rapamycin encapsulated in Rap-PLGA NPs was sufficient to obtain the desired effect at low doses (see Section 3.3).

#### 3.1.5. Kinetic Analysis of Rapamycin Release

The kinetics of rapamycin release from NPs was determined by microdialysis method using a side-by-side diffusion bag. (Figure 3). The rapamycin release displayed a biphasic profile: a burst effect occurred during the first three hours due to polymer erosion and further diffusion of the drug, followed by a delayed and extended drug release from 3 h to 160 h (Figure 3). These data were analyzed using various mathematical models (Table 2), with the Korsmeyer–Peppas model proving most effective. This model is a valuable tool in pharmaceutics for understanding drug release mechanisms, allowing to interpret the release process by examining the coefficients within the equation. In this case, the rapamycin release data implied a diffusion-controlled drug release [68]. Other drugs loaded in polymeric systems followed the same kinetic behavior [30,69,70]. However, the burst effect phase of the rapamycin release was better characterized by a zero-order model, which implies that the release of drug occurred at a constant rate and did not depend on the concentration but on the elapsed time of release. Due to the controlled release capacity, protection of macromolecules from degradation and unneeded efflux, receptor-mediated transport is one of the most promising brain drug delivery strategies.

### 3.2. Biological Characterization of NPs

#### 3.2.1. Uptake of NPs by Neuronal and Glial Cells

The SCA7-associated degenerative process mainly affects the cerebellum, brainstem, and spinal cord [8,71,72], with the retinal ganglion, and visual cortex being impaired as well [73,74,75]. Consistently, aggregates of mutant ataxin-7 have been found in glial cells, Purkinje cells and hippocampal CA1 pyramidal neurons [76,77,78]. At the retina, inclusions of ataxin-7 appear first in Müller glia and cones and interneurons of the outer and inner nuclear layers, respectively, and later in the rods and ganglion cells [76,79]. Therefore, we planned to evaluate our delivery nanosystem in two cell lines derived from the CNS, namely MIO-M1 (Müller glial) and SH-SY5Y (neuronal) cells. We proceeded to encapsulate the fluorescent probe coumarin-6 (C6) into PLGA NPs, to evaluate the internalization of NPs into Müller glial and neuronal cells using fluorescent microscopy. The absence of fluorescence in cells incubated with PLGA NPs alone indicate that NPs are not self-fluorescent (Figure 4). On the contrary, the intense fluorescent signal found in the cytoplasm of both cell types, after two hours of incubation with C6-PLGA NPs implies an efficient uptake of NPs (Figure 4). Clearly, penetration of C6 into SH-SY5Y was most efficient when encapsulated into NPs, compare with C6 alone. The efficient uptake of NPs by neuronal and glial cells could be based on their relevant functional features. Firstly, PLGA improves cell internalization via clathrin-mediated endocytosis [80,81], the endocytosis route present in the CNS. Secondly, the nano-size, spherical shape, and surface characteristics of NPs enable them to be uniformly internalized into cells [43,82]. Lastly, the efflux-pump inhibitory activity and neuroprotective effect of Poloxamer 188 [83] confer additional beneficial characteristics on NPs. Furthermore, the receptor-mediated endocytosis of NPs is enhanced by Poloxamer 188 through its ability to selectively adsorb apolipoproteins E and B [84,85].

#### 3.2.2. Effect of NPs on Cell Viability and Morphology

Because of their small size, NPs might disrupt the normal molecular/biochemical environment of the cell and damage sensitive cellular structures. For instance, titanium dioxide NPs elicited oxidative stress and mitochondrial dysfunction in rat (C6) and human (U373) glial cells [86]. Thus, to evaluate the potential toxicity of Rap-PLGA NPs, a cell viability assessment was carried out using MTT assays. To this end, MIO-M1 and SH-SY5Y cells were incubated for 24 h or 72 h with different concentrations (3-1500 µg/mL) of empty and rapamycin-loaded NPs, prior to be lysed and subjected to cell viability quantification. The use of the empty PLGA NPs in glial cells decreased the viability of MIO-M1 cells at the highest doses used (≥500 mg/mL) (Figure 5A). A similar decrease in cell viability was specifically observed in neuronal cells treated for 72 h with doses of empty PLGA NPs ranging from 200 to 1500 mg/mL (Figure 5C). Interestingly, there was no effect on cell viability in any of the cell lines when treated with the different concentrations of Rap-PLGA NPs at the two different incubation times (Figure 5B,D). We hypothesize that the rapamycin incorporated within the NPs had a protective antioxidant effect against the change in pH that occurred during the hydrolysis of NPs [87,88]. Conversely, incubation with the cell death activators staurosporine and H_2_O_2_ (positive controls) resulted in a marked decrease in cell viability in both cell lines, confirming the sensitivity of MTT assays.

Next, we evaluated whether treatment with NPs exerts any negative effect on cell structures. MIO-M1 and SH-SY5Y cells were incubated for 72 h with 25 µg/mL or 150 µg/mL of PLGA NPs and Rap-PLGA NPs, prior to confocal microscopy analysis. No significant alterations were observed for both cellular and nuclear morphologies, as shown by the morphometric analyses of phalloidin-staining cells and DAPI-stained nuclei, respectively (Figure 6 and Appendix A). Overall, these data confirmed that treatment with NPs did not cause any apparent damage in cell viability or integrity *in vitro*.

#### 3.2.3. Analysis of NPs Cytotoxicity in Glial Cells

Glial cells play a key role in brain homeostasis through a neurovascular and neurometabolic coupling with neurons [89]; in SCA7, it is thought that glial cell proliferation might enhance vascular and metabolic activities in response to the neuronal loss observed in SCA7 [90]. With respect to the retina, it has been shown that the SAGA deubiquitinase activity is required in glial cells rather than neurons to drive neural connectivity and axon guidance during the visual development of Drosophila [91]. Furthermore, expression of mutant ataxin-7 in glia is sufficient to induce neurodegeneration in a SCA7 mouse model [92]. Collectively, this evidence highlights the glial contribution to the pathogenesis of SCA7. Therefore, we were prompted to analyze in depth the safety of our delivery nanosystem in cultured glial cells. To this end, we ascertained whether treatment with empty or rapamycin-loaded NPs activates apoptosis in MIO M1 cells by Annexin V/IP and caspase-3 and -9 activation assays. MIO M1 cells treated with staurosporine, a well-known activator of apoptosis (positive control) induced late apoptosis in 50% of MIO M1 cell cultures, as shown by flow cytometry analysis, as well as activation of apoptotic caspase-3 and -9, evidenced by the presence of their cleaved dimmer fragments in the immunoblots (Figure 7A,B). In the opposite way, a subtle non-significant increase in early apoptotic cells (~10–12%) and virtually non-late apoptotic/death cells were found in cells treated with 25 µg/mL or 150 µg/mL of PLGA NPs and Rap-PLGA NPs (Figure 7A). Likewise, none of these treatments provoked activation of apoptotic caspsase-3 and -9 in MIO M1 cells (Figure 7B). Overall, these results demonstrate both the suitability and safety of our nanocarrier system to target neuronal and glial cells *in vitro* and deliver rapamycin within them.

### 3.3. Application of Rapamycin-Loaded NPs to Alleviate the SCA7 Phenotype in a Glial Cell Model

Rapamycin can clear toxic polyQ and polyalanine-expanded proteins by inducing autophagy, reducing thereby their toxicity [93]. Specifically, it has been shown that treatment with rapamycin reversed the accumulation of polyQ huntingtin, which consequently resulted in the amelioration of cell death in cellular models of Huntington disease (HD); furthermore, a neuroprotective effect for this therapy was demonstrated in a *Drosophila* model of HD [40,41] Remarkably, rapamycin drove the elimination of truncated fragments of mutant ataxin-7 carrying a polyglutamine track of 65 residues, in an inducible PC12 cell model, which proved the therapeutic potential of inducing autophagy to treat SCA7 [94]. Nonetheless, the therapeutic effectivity of rapamycin is limited *in vivo* by the BBB, which significantly affects its bioavailability/biodistribution [31,38,42,95]. To overcome this obstacle, we provided herein an efficient/safe nano-delivery system to treat SCA7.

#### 3.3.1. Treatment with Rap-PLGA NPs Promotes the Clearance of Mutant Ataxin-7 Aggregates by Inducing Autophagy in a SCA7 Glial Cell Model

The therapeutic potential of Rap-PLGA NPs was evaluated in a MIO-M1 cell-based inducible model of SCA7. We previously generated a MIO-M1 derivative cell line that expresses human ataxin-7 bearing a pathological tract of 64 glutamine (Q64) residues, under the Tet-on inducible expression system control (MIO-M1-Q64 cells) [34]. A calibration curve of doxycycline was made (Appendix A) and 1 μg/mL of doxycycline was selected for the induction of the system. Upon induction with doxycycline for 24 h, Q64 ataxin-7 was found to localize in the nucleus of MIO-M1-Q64 cells, and to have the ability to form protein aggregates (a main SCA7 cellular hallmark), as shown by confocal microscope analysis of immunostained cells using ataxin-7 antibodies (Figure 8A). Remarkably, treatment with Rap-PLGA NPs and rapamycin alone, but not PLGA NPs, resulted in a marked decrease in Q64 ataxin-7 aggregates per cell (Figure 8A), which implies that both treatments had the ability to clear these toxic protein bodies. To demonstrate that the removal of protein aggregates occurred through autophagy induction, MIO-M1-Q64 cells expressing Q64 ataxin-7 were treated for 24 h with PLGA NPs, Rap-PLGA NPs, or rapamycin alone, and chloroquine (CQ; inhibitor of the fusion between autophagosomes and lysosomes) at the same time. The use of CQ allowed us to stop autophagy and evaluate autophagic activity by the accumulation of autophagosomes, which were immunostained using antibodies against the autophagosomal protein marker microtubule-associated protein 1A/1B-light chain 3A (LC3) [96], prior to be analyzed by confocal microscopy (Figure 8B). As expected, an increased number of LC3 foci were observed in cells treated with Rap-PLGA NPs or rapamycin alone, compared to cells treated with PLGA NPs (Figure 8B), which implies that clearance of Q64 ataxin-7 aggregates in MIO-M1-Q64 cells was conducted indeed by autophagy activation.

Interestingly, the treatment with rapamycin alone caused a notable cellular stress, in opposite to Rap-PLGA NPs (Appendix A), which highlights the advantage of the nanoparticulate system in releasing the drug in a controlled manner. Therefore, the polymeric nanocarrier system is relevant to prolong drug’s half-life, avoiding its premature metabolism, and reducing the cellular toxicity by having prolonged drug liberation, which ultimately appears to optimize the removal of mutant ataxin-7 aggregates ion (mATXN7) in the cellular model of SCA7.

#### 3.3.2. Treatment with Rap-PLGA NPs Prevented Apoptosis in a SCA7 Glial Cell Model

The presence of mutant ataxin-7 aggregates has been mechanistically linked to neuronal/glial cell death; specifically, neuronal vulnerability, retinal degeneration and/or rod receptors apoptotic death were reported in SCA7 mouse models [76,97,98]. Furthermore, the expression of a polyQ ataxin-7 induced apoptotic cell death in cultured cerebellar neurons via activation of caspase-3 and caspase-9 [99]. Thus, we anticipated that the expression of Q64 ataxin-7 might elicit apoptosis in MIO-M1-Q64 cells, and that the clearance of ataxin-7 protein aggregates mediated by Rap-PLGA NPs might be reflected in apoptosis alleviation. Supporting our hypotheses, activation of apoptosis was found upon induction of Q64 ataxin-7 expression, as shown by the presence of caspase-9 cleaved fragments in the immunoblots of Dox-treated MIO-M1-Q64 cells (Figure 8C). On the other hand, a slight induction of the caspase-3 cleaved fragment was found upon DOX induction in both MIO-M1-Q10 and MIO-M1-Q64 cells, indicating that such induction was irrespective of mutant ataxin-7 expression.

Moreover, a significant decrease in the levels of the proteolytic fragments for both caspases was observed upon incubation of the induced MIO-M1-Q64 cells with Rap-PLGA NPs (Figure 8C); this effect can reasonably be attributed to the presence of encapsulated rapamycin. The beneficial effect of PLGA NPs with the proteolytic fragment caspase-9 could be attributed to Poloxamer 188, because this polymer exerts a cytoprotective effect by restoring the lysosomal membrane integrity and preventing thereby the caspase-dependent apoptotic cell death [100]. Collectively our results integrate a proof of concept of the therapeutic potential of Rap-PLGA NPs in the SCA7 glial cell model, namely clearance of Q64 ataxin-7 aggregates, and the subsequent prevention of apoptosis. However, further studies using other CNS cell types, such as neurons, are required to strengthen the therapeutic impact of our nanosystem. 

Finally, it is worth mentioning that the effectiveness of our nanocarrier system needs to be evaluated *in vivo*. Previous reports have shown marked differences in the internalization of nanocarriers between *in vitro* and *in vivo* systems, primarily attributed to the presence of natural barriers within an organism [101]. Therefore, the first challenge is the limited penetration of NPs across the BBB. Typically, high dosages of NPs are required to facilitate their passive crossing of the blood–tissue interface. Nonetheless, passive accumulation of NPs does not guarantee delivery into the target tissue, decreasing their therapeutic effectiveness and can potentially lead to side effects, including immunotoxicity [102]. Thus, the physicochemical properties of NPs are essential to obtain proper bioavailability and biodistribution, which in turn will define the administration route and treatment frequency [103]. Specifically, since rapamycin is an insoluble molecule, the nanoparticulate system designed here enhances its bioavailability and biodistribution because surfactants like poloxamer 188 and PVA provide solubility and stability, which ultimately would facilitate drug release in a prolonged/modulated manner.

## 4. Conclusions

We developed a nanocarrier system (Rap-PLGA NPs) with physicochemical and biological characteristics relevant for therapeutic use against SCA7 and other polyQ diseases. Rap-PLGA NPs were well tolerated by neuronal and glial cell lines showing no adverse effects on cell viability or cellular structures. Noteworthy, Rap-PLGA NPs could clear toxic aggregates of mutant ataxin-7 by inducing autophagy in a SCA7 glial cell model. Future evaluation in an SCA7 mouse model is required to test the ability of this nanocarrier system to cross the BBB, reach the CNS, and exert its therapeutic effect. 

## Figures and Tables

**Figure 1 cells-12-02735-f001:**
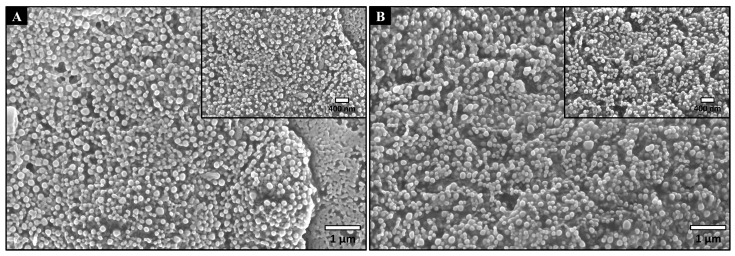
Micrographs of NPs obtained by SEM. (**A**) PLGA NPs; magnification, 25,000× and 40,000×. (**B**) Rap-PLGA NPs; magnification, 25,000× and 40,000×.

**Figure 2 cells-12-02735-f002:**
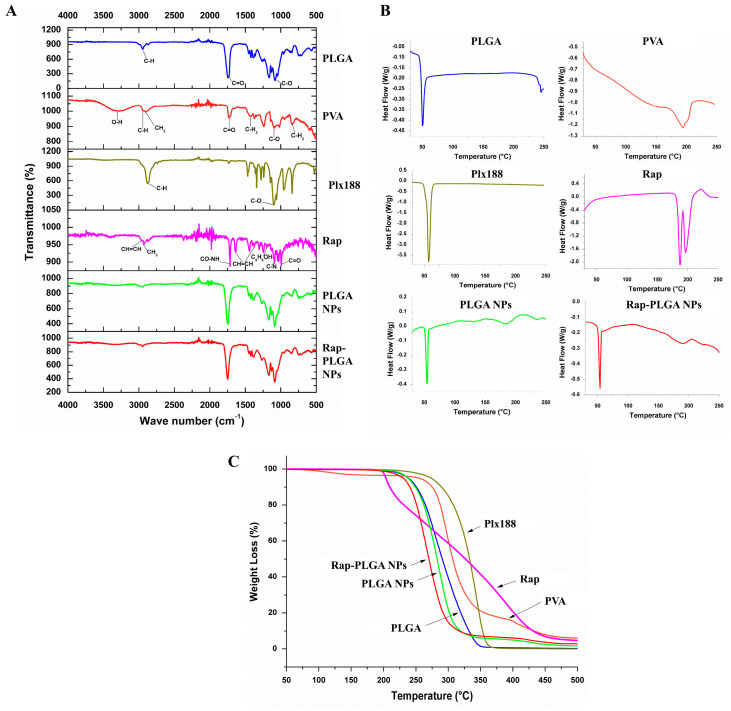
Structural and thermal characterization of PLGA NPs and Rap-PLGA NPs. (**A**) Fourier-transform infrared spectroscopy (FTIR) analysis of raw excipients and NPs formulations; (**B**) Differential scanning calorimetry (DSC) of raw excipients and NPs formulations; (**C**) Thermogravimetric analysis (TGA) of raw excipients and NPs formulations.

**Figure 3 cells-12-02735-f003:**
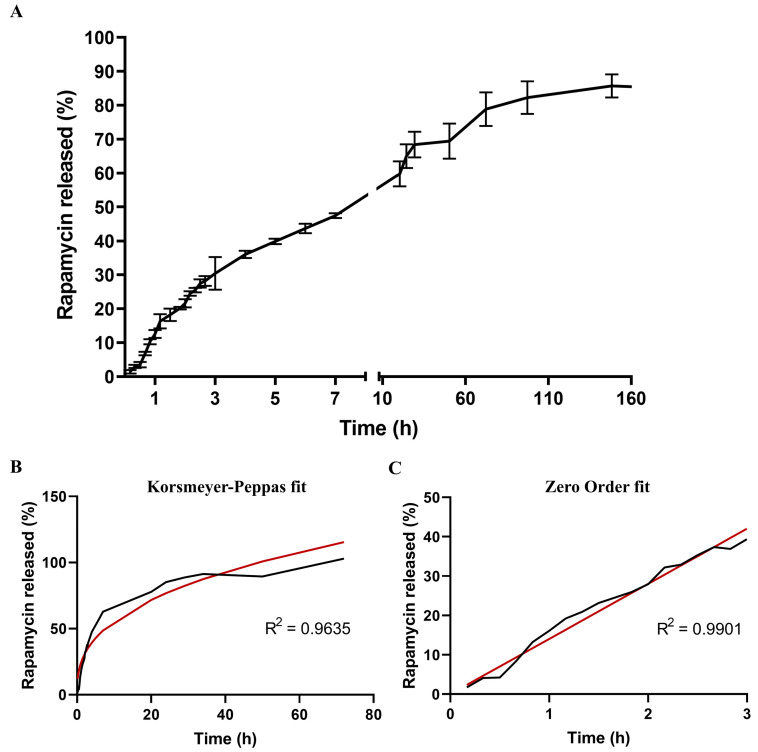
Release profile of rapamycin from Rap-PLGA NPs. (**A**) Time course of rapamycin release from Rap-PLGA NPs using the microdialysis method. Error bars represent standard deviations from triplicate experiments; (**B**) Rapamycin release profile adjusted to the Korsmeyer–Peppas model. (**C**) Rapamycin release profile during the burst effect fitted to the zero-order model. The experimental data (black line) is compared with the theoretical release profile (red line) predicted by the Korsmeyer-Peppas equation (controlled drug delivery systems) and to the zero-order model (burst phenomenon), respectively.

**Figure 4 cells-12-02735-f004:**
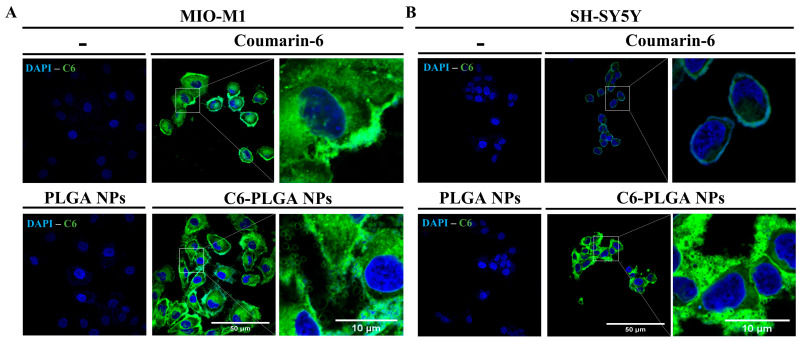
Internalization of NPs in glial and neuronal cells. Cells grown on coverslips were incubated for 2.5 h with 150 μg/mL of C6-PLGA NPs or coumarin-6 (0.18 µg) alone. Then, cells were stained with DAPI to visualize nuclei before being analyzed by confocal microscopy. (**A**) MIO-M1 cells. (**B**) SH-SY5Y cells. Representative images are shown.

**Figure 5 cells-12-02735-f005:**
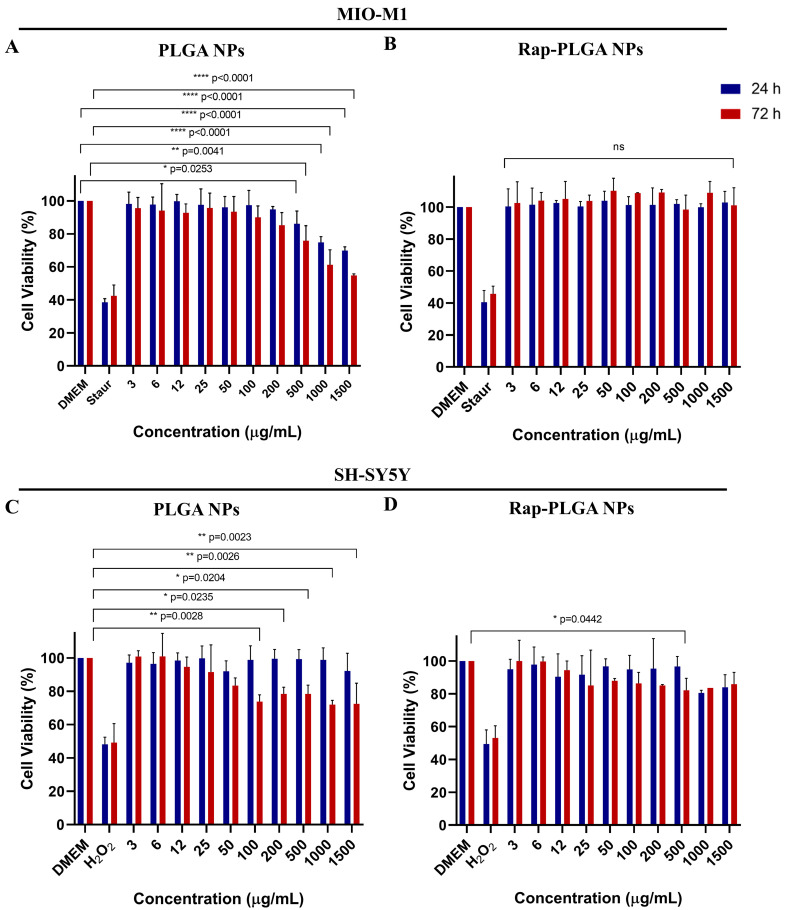
Analysis of cell viability in neuronal and glial cells treated with PLGA NPs or Rap-PLGA NPs. MIO-M1 cells were treated for 24 h or 72 h with the indicated amounts of PLGA NPs (**A**) or Rap-PLGA NPs (**B**), prior to being subject to MTT-based cell viability assays. Cells were treated for 4 h with 2 µg/mL of staurosporine (Staur) as a positive control for cell death. SH-SY5 cells were treated for 24 h or 72 h with the indicated amounts of PLGA NPs (**C**) or Rap-PLGA NPs (**D**) and then subjected to MTT-based cell viability assays. Cells were treated with 2% H_2_O_2_ for 4 h as a positive control for cell death. Data correspond to three separate experiments with significant differences obtained by one way-ANOVA with Dunnett correction for multiple comparisons; each condition was compared to the DMEM condition. Cell viability data corresponding to the control condition (untreated cells grown in DMEM) were set up at 100% viability for comparison. Bars indicate mean ± SEM. * Statistically significant difference; ns: non-significant differences.

**Figure 6 cells-12-02735-f006:**
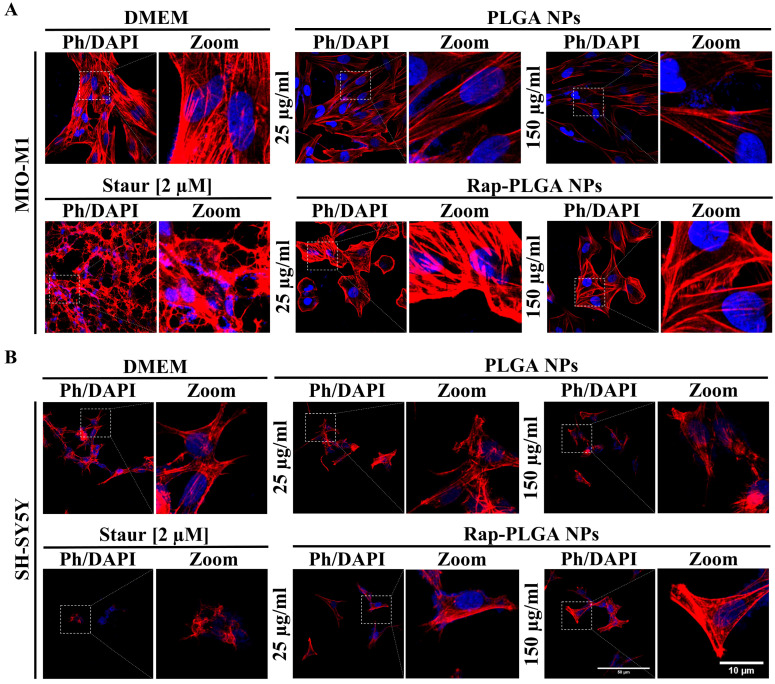
Analysis of cell morphology in neuronal and glial cells treated with PLGA NPs or Rap-PLGA NPs. MIO-M1 (**A**) and SH-SY5Y (**B**) cells were incubated for 72 h with 25 µg/mL or 150 µg/mL of PLGA NPs or Rap-PLGA NPs, prior to being stained with rhodamine-conjugated phalloidin (Ph) and DAPI to decorate the actin-based cytoskeleton and nuclei, respectively. Cells preparations were then subjected to Confocal Laser Scanning Microscopy analysis.

**Figure 7 cells-12-02735-f007:**
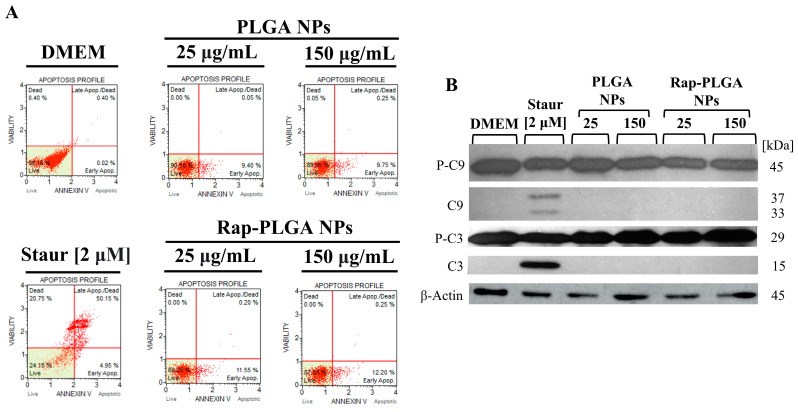
Cytotoxic characterization of PLGA NPs or Rap-PLGA NPs in MIO M1 glial cells. (**A**) Assessment of early and late apoptosis by cytometry flow in MIO-M1 cells treated for 72 h with the indicated concentrations of each NPs formulation. Treatment with 2 µM staurosporine (Staur) was used as an inducer of apoptosis (positive control). (**B**) Analysis of caspase-3 and caspase-9 activation in MIO-M1 cells treated as per A. Cells treated with 2 µM staurosporine (Staur) were used as a positive control for apoptosis induction. Cell lysates were subjected to immunoblotting assays using antibodies against caspase-3, caspase-9, and *β-*actin (loading control). Pro-caspase 9 (P-C9), caspase-9 (C9), pro-caspase 3 (P-C3), and caspase-3 (C3). Data correspond to representative images of each assay.

**Figure 8 cells-12-02735-f008:**
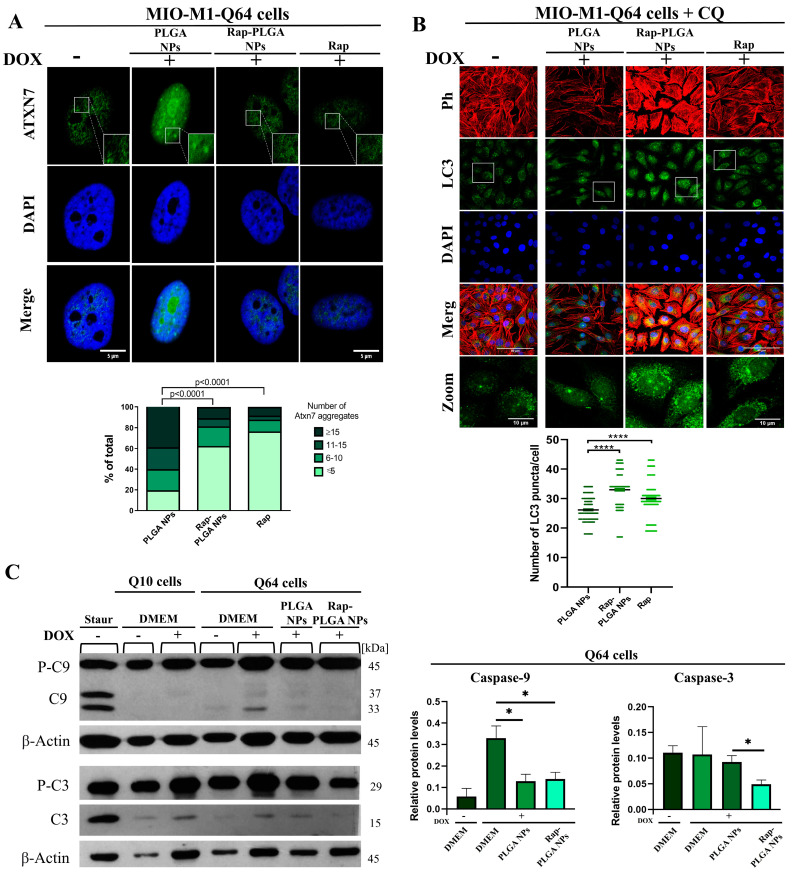
Therapeutic efficacy of Rap-PLGA NPs in a SCA7 glial cell model. (**A**) MIO-M1-Q64 cells grown on coverslips were incubated with doxycycline (Dox) for 24 h to induce Q64 ataxin-7 expression and were then treated with the indicated NPs or rapamycin alone, prior to being immunolabeled with antibodies against ataxin-7 and analyzed by confocal microscopy. Typical images from three independent experiments are shown; bar, 5 µm. *Down.* The percentage of cells with Q64 ataxin-7 aggregates, and the number of aggregates per cell were quantified manually using Image J (n > 30 cells per condition from three independent experiments), with significant differences determined by χ^2^-test. (**B**) MIO-M1-Q64 cells grown on coverslips were induced to express Q64 ataxin-7 as per A and were then treated for 24 h with the indicated NPs or rapamycin alone and chloroquine (CQ) at the same time. After the treatment, the cells were immunostained for LC3 and counterstained with DAPI and rhodamine-conjugated phalloidin to decorate nuclei and the actin-based cytoskeleton, respectively, prior to being subjected to confocal microscopy analysis. The white square highlights the zoomed-in area. Representative images from three separate experiments are shown; bar, 10 µm. *Down.* The number of LC3 punctate per cell was quantified manually using Image J (n > 80 cells per condition from three independent experiments), with significant differences determined by the Kruskal–Wallis test and Dunn’s as a post hoc test (**** *p* < 0.0001). (**C**) To analyze apoptosis, lysates from cell cultures incubated with Dox to express Q10 ataxin-7 or Q64 ataxin-7 were further treated for 24 h with PLGA NPs or Rap-PLGA NPs and subjected to Western blotting using antibodies against caspase-3, caspase-9, and actin (loading control). Representative image from three separate experiments is shown. *Right*. Relative caspase-3 or caspase-9 protein levels after Dox induction in Q64 cells and treatment with the indicated NPs. Statistical differences with significant differences determined by *t* tests using the Holm–Sidak correction; * = 0.05.

**Table 1 cells-12-02735-t001:** Characterization of PLGA NPs and Rap-PLGA NPs.

Sample	Size (nm) ± SD	PDI ± SD	Z-Potential (mV) ± SD
PLGA NPs (lyophilized)	212.73 ± 4.87	0.053 ± 0.02	−17.66 ± 0.40
Rap-PLGA NPs (lyophilized)	211.86 ± 4.037	0.045 ± 0.017	−18.53 ± 0.288
Stability Assay (PLGA NPs dispersed)			
Time of storage	Size (nm) ± SD	PDI ± SD	Z-Potential (mV) ± SD
1 month	226.5 ± 5.074	0.031 ± 0.03	−17.9 ± 0.436
2 months	221 ± 7.016	0.048 ± 0.031	−20.6 ± 0.503
3 months	217.2 ± 4.359	0.079 ± 0.037	−19.3 ± 0.755
4 months	208.7 ± 2.868	0.041 ± 0.007	−13.6 ± 0.557
6 months	205.7 ± 2.723	0.037 ± 0.019	−14 ± 0.379

**Table 2 cells-12-02735-t002:** Rapamycin release profile adjusted to mathematical models.

**Rapamycin release profile (entire time range)**				
**Mathematical model**	**Equation**	**R^2^**	**K**	**n**
Zero order	Q_t_ = Q_0_ + K_0_t	0.83	2.13	0
First Order	ln Q_t_ = lnQ_0_ + K_1_t	0.52	0.066	0
Higuchi	Q_t_ = K_H_t^1/2^	0.94	15.60	0
Korsmeyer–Peppas	Q_t_/Q_∞_ = K_k_t_n_	0.96	23.57	0.37
**Rapamycin release profile (burst effect)**				
**Mathematical model**	**Equation**	**R^2^**	**K**	**n**
Zero order	Q_t_ = Q_0_ + K_0_t	0.99	14.01	0
First Order	ln Q_t_ = lnQ_0_ + K_1_t	0.85	1.32	0
Higuchi	Q_t_ = K_H_t^1/2^	0.99	19.89	0
Korsmeyer–Peppas	Q_t_/Q_∞_ = K_k_t_n_	0.99	15.05	0.91

Q_t_: amount of drug released in time t; Q_0_: initial amount of drug in the dosage form; Q_∞_: total amount of drug dissolved when the dosage form is exhausted; K_0_, K_1_, K_H_, K_s_, K_k_: release rate constants; n: release exponent (indicative of drug release mechanism).

## Data Availability

MIO-M1 retinal glial cells (accession number: KX350100/European Nucleotide Archive) was a donation directly from Moorfields/Institute of Ophtalmology at University College of London (Stephen Moss) and SH-SY5Y human neuroblastoma cells (accession number: PRJNA723610/National Center for Biotechnology Information) was obtained by CRL-266^TM^ (ATCC).

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
