# Peer review of "Development of a Polymeric Pharmacological Nanocarrier System as a Potential Therapy for Spinocerebellar Ataxia Type 7"

_cells, 2023, doi:10.3390/cells12232735_

Round 1

Reviewer 1 Report

Comments and Suggestions for Authors

1. The study is lacking in terms of accessibility of the rapamycin-loaded NPs through the Blood brain barrier in an animal model.  While the current paper is missing these critical experiments, the authors have acknowledged this weakness and stated they will follow up on this in upcoming work with in vivo studies in SCA7 mouse models, which is acceptable.

2.  Authors have only conducted biological characterization such as uptake, cell viability assay and morphology assays in vitro.  It is encouraged that the authors visit these parameters through in vivo treatments of the Rap-PLGA-NPs. In vitro kinetics can be very different from in vivo pharmacokinetics of the NP system.  

3. Authors have only demonstrated NP cytotoxicity data in MIO M1 cells and functional data with respect to clearance of toxic aggregates and apoptosis prevention data in an inducible SCA7 retinal glial cell model MIO-M1-Q64.  Authors should include additional cell lines to demonstrate in vitro functional efficacy of Rap-PLGA-NPs.  Can authors demonstrate efficacy of the treatment in additional SCA7 phenotype presenting cell models?

4. Authors must include some discussion on how each set of the in vitro data outlined in this manuscript will likely translate in vivo and what are the challenges of in vitro to in vivo translatability with this system besides BBB access.

5. Minor: Elaborate on the full-form of the abbreviation 'SNC' on Line 619.

Comments on the Quality of English Language

No issues with English language and phrases.

Reviewer 2 Report

Comments and Suggestions for Authors

In their presented work, Borbolla-Jiménez and colleagues developed a nanoparticle drug carrier, thoroughly tested its physicochemical properties, and utilized it to analyze the biocompatibility and clearance of polyQ-ataxin-7 insoluble aggregates. Their PLGA NPs, as well as rapamycin-loaded Rap-PLGA NPs, exhibited appropriate size, PDI, and electrical potential for drug carriers. The nanoparticles did not show enhanced toxicity in cell culture, and Rap-PLGA NPs appeared to clear aggregated polyQ protein through an autophagy process.

Drug delivery to the central nervous system presents a significant challenge, and both the neurobiology and pharmacology fields are eagerly anticipating a breakthrough technology that can safely treat patients with neurodegenerative conditions. The work by Borbolla-Jiménez and colleagues is highly anticipated and represents a crucial step towards achieving this goal. The authors' approach to creating a nanoparticle drug carrier and demonstrating the concept within the context of SCA7 is methodologically sound and well-communicated. The literature is appropriately cited. However, the manuscript would benefit from some clarifications and improvements.

  1. In Figure 8C, the authors demonstrate the impact of rapamycin-loaded NPs on apoptosis by measuring caspases 3 and 9 cleavage as a readout. An appropriate positive control, staurosporine in wild-type cells, is used. For MIO-M1-Q64 cells, the authors employ a -dox control and compare it to the +dox treated cells, assuming that the increase in apoptosis results from dox-induced ataxin-7 expression. However, it is worth noting that doxycycline alone is known to increase apoptosis/caspase response in some cell lines. Therefore, the authors should consider adding wild-type cells treated with doxycycline to control for such a possibility.

  1. Table 2. I suggest including a figure alongside table 2 that illustrates the model's fit to the rapamycin release data. The authors can display both the Korsmeyer-Peppas fit for the entire range and the zero-order fit for the burst. This would provide valuable visual assistance to the reader in interpreting the data. Additionally, the table should have an upper label similar to the 'Rapamycin release profile (burst effect)' label, which could read 'Rapamycin release profile (entire time range)' or a similar alternative.

  1. Figure 6 and the associated text: The analysis of cell and nucleus morphology should be quantitative. Metrics such as cell footprint, nuclear size, shape, and intensity can be readily calculated from existing image data using open-source platforms like ImageJ or Cell Profiler.

  1. In Figure 5, the communication of data analysis is not entirely clear. If I understand correctly, even though the chart displays both 24 and 72-hour treatment times, it seems that they are not analyzed together, which justifies the use of a one-way ANOVA. However, if the intention is to analyze the data together, a two-way ANOVA would be more suitable, considering treatment time and drug concentration as independent variables. Could the authors clarify what post-test comparison method was employed to identify the source of the observed differences? Was it used to compare each concentration point to the DMEM condition? Moreover, it appears that the DMEM condition is lacking error bars. This omission requires clarification. Ideally, for each treatment time, a one-way ANOVA with Dunnett correction for multiple comparisons, comparing each concentration to the DMEM condition, would be employed.

  1. Figure 8A: Is the image of a PLGA NPs cell representative? The staining seems qualitatively different from the other conditions. Could this be attributed to the effect of dox-induced ataxin-7 expression? Furthermore, the foci detection may not be entirely convincing based on the provided images. Were they counted manually in ImageJ or using a dedicated plugin? To enhance the reader's experience, I recommend including higher resolution images that allow for zooming in to observe finer details.

  1. The supplementary figures lack labels and captions. This needs to be corrected.

Minor issues:

  1. In line 63, the authors state that current therapeutic approaches primarily focus on proteolytically eliminating mutant ataxin-7 aggregates. While this approach has been extensively explored, it is important for the authors to acknowledge that interventions aimed at regulating ataxin-7 expression through ASOs and RNAi mediators have emerged as strong potential candidates for SCA7 and other polyQ diseases therapy.

  1. While the Methods section specifies the use of standard deviation after the measurements in Table 1, it would be beneficial to include this information in the top row of the table. Additionally, it is advisable to use the '+'/-' sign instead of just '+' to indicate both positive and negative values.

  1. Figure 2: Readers would benefit from having the legend right next to the plots instead of encoding components with letters and then decoding them in the figure caption.

  1. Caption of figure 4 (line 453). The use of the term 'Magnification, 5×' is unfortunate. While I understand that it represents a relative magnification increase, it may be more suitable to use absolute magnification values instead.

  1. Authors should expand the use of the term 'confocal laser scanning microscopy' instead of using the abbreviation CLSM, especially since they use it only once (line 495).

Reviewer 3 Report

Comments and Suggestions for Authors

The paper is well and clearly writen. In my opinion, it should ben stated from the intoduction on, that the study was carried out in cultured cells as preliminary step to design animals experiments.

Author Response

Thank you for your feedback. In the revised version of the paper, we have incorporated a discussion from the introduction onward to clearly indicate that the study was initially conducted in cultured cells as a preliminary step for designing animal experiments.

Round 2

Reviewer 1 Report

Comments and Suggestions for Authors

The authors have addressed comments from Report 1 and adequately incorporated edits in the manuscript.  Primarily, authors have acknowledged areas in which their study is lacking and further investigation is needed to evaluate the nanocarrier as a therapeutic vehicle.  

This study shows the therapeutic potential of a novel nanocarrier loaded with Rapamycin for the treatment of Spinocerebellar Ataxia Type 7 and is a valuable avenue for further development.  

Reviewer 2 Report

Comments and Suggestions for Authors

The authors addressed all the questions and clarification needs. I recommend the paper for publication

Reviewer 3 Report

Comments and Suggestions for Authors

My suggestion has been addedm through the text. Accept at it is.